# The Positive Impact and Associated Mechanisms of Physical Activity on Mental Health in Underprivileged Children and Adolescents: An Integrative Review

**DOI:** 10.3390/bs10110171

**Published:** 2020-11-08

**Authors:** Lauren T. Rose, Andrew Soundy

**Affiliations:** School of Sport, Exercise and Rehabilitation Sciences, University of Birmingham, Edgbaston, Birmingham B15 2TT, UK; a.a.soundy@bham.ac.uk

**Keywords:** physical activity, mental health, childhood, adolescence and underprivileged

## Abstract

(1) Background: Review-based studies are required to consider the different designs and data that describe the association between physical activity and mental health for underprivileged children and adolescents. There is a particular need to identify mechanisms which could explain the association, and factors which influence the association in this population group. (2) Methods: An integrative review with a systematic search was conducted in three stages: (i) a systematic literature search on four databases from inception until May 2020. Studies were eligible if they examined underprivileged children, involved moderate to vigorous physical activity (MVPA) and had a focus on at least one outcome measure related to mental health. (ii) Two critical appraisal tools were used to assess quality across different study designs. (iii) Synthesis was undertaken in four stages, examining social, internal and physical outcomes and mechanisms. (3) Results: A total of 16 studies were included (2 qualitative; 12 quantitative and 2 mixed methods). Fifteen (15/16; 93.8%) showed an association between MVPA and a significant improvement in at least one mental health outcome. The most studied outcomes included: internal aspects (*n =* 14), significant effects were identified in 12/14 studies; social aspects (*n =* 6), significant effects were identified across all studies and finally physical aspects (*n =* 5), significant effects were found in 3/5 studies. Autonomy support was identified as a mechanism to explain the change, whilst age and gender influenced the strength of the association. (4) Conclusions: The findings provide confirmation of the link between MVPA and the improved mental health and wellbeing of disadvantaged children and adolescents. Future research must consider the long-term effect through longitudinal studies, along with determining whether any specific types of PA are more impactful than others.

## 1. Introduction

It is estimated that 10–20% of children and adolescents worldwide are living with a mental health problem [1]. For instance, 14% of Australians aged between 4 and 17 were diagnosed with a mental health disorder in 2013–2014 [2]. Evidence has also identified an increased trend over time. For instance, one British study [3] identified an increase in the prevalence of mental health conditions between 1986–2006 of 8.3% to 12.8% in boys and 16.4% to 28.4% in girls. It is of the utmost importance that research addresses mental health in children and adolescents since it is estimated that around half of young adults who have a mental health problem also experienced one as a child [4]. Adolescents with a mental health disorder are identified as isolated, find it difficult to regulate their emotions and have less social support than their age-matched peers [5]. For the purpose of this article, the following definition of mental health will be taken:


*‘a dynamic state of internal equilibrium which enables individuals to use their abilities in harmony with universal values of society. Basic cognitive and social skills; ability to recognize, express and modulate one’s own emotions, as well as empathize with others; flexibility and ability to cope with adverse life events and function in social roles; and harmonious relationship between body and mind represent important components of mental health which contribute, to varying degrees, to the state of internal equilibrium’*
[6] (pp. 231–232)

Physical activity is a broad term which describes the expenditure of energy due to the contraction and extension of skeletal muscles, resulting in bodily movement [7]. It is recommended that children and adolescents complete 60 min of moderate to vigorous physical activity (MVPA) daily, whilst incorporating muscle and bone-strengthening exercises 3 times a week [8]. A global survey of 1.6 million adolescents, conducted in 2016, stated that 81% of the sample were insufficiently physically active [9]. A systematic review of longitudinal studies which examined the change in PA between ages of 10 and 19, found that on average, there was a 7% decrease in PA every year [10]. Multiple factors have been identified as influencing the decline including, the attraction of computer games, social media and parental safety concerns with playing outside [11]. However, many schools are also significantly reducing the number of hours allocated to physical education e.g., [12,13,14]. Research has shown that physical activity is beneficial for both the physical, e.g., [9,15], and mental health, e.g., [16,17,18], of children and adolescents. Research considering MVPA support these findings, e.g., [19,20,21,22,23], and there is also evidence for a reduction in the likelihood of developing mental health problems for adolescents who are more physically active [16,24]. Research which specifically considers underprivileged children (there is no conclusive definition of underprivilege so for the purpose of this investigation any study which uses the terms: low SES; low income; socioeconomically disadvantaged; disadvantaged or underprivileged, when describing the sample population will be included) is more limited. Physical activity levels are known to be significantly lower amongst people from low socioeconomic backgrounds, e.g., [25,26], and gender differences have been reported with girls being more likely to be inactive (RRs ranging from 1.04 to 1.12, *p* <  0.01) [27]. Additionally, socially disadvantaged children and adolescents are significantly more likely to suffer from mental health problems (OR 4.16, 95% CI 2.49–6.94) [28].

There is an urgent need for review-based research to consider the impact of PA on the mental health of this cohort. Such a review would need to address specific considerations: (a) the impact of various settings of study, for instance, past research has considered children who live in socially disadvantaged areas [29], have low family incomes [30], have been identified as being ‘at risk’ of social disengagement [31], or who attend a school in a socially disadvantaged area [32]. However, current review evidence has only considered out-of-home care settings [33]; (b) the impact on wide ranging mental health outcomes including self-esteem, anxiety and depression [34]. To date, no reviews have been conducted which specifically examine the impact on the mental health of disadvantaged children and adolescents; (c) the factors or mechanisms that could influence change in mental health outcomes, e.g., age [30], and autonomy support, e.g., [35]. To the best of the authors’ knowledge, no review evidence has explored this previously; and, (d) the ability to incorporate both qualitative based research, e.g., interviews [36,37], and quantitative based research, e.g., questionnaires [30,38]. One design that would meet this need is the integrative review [39]. This type of review will provide a more in-depth analysis of current and past policy development, the realities of practice and research coverage within the field.

Aim: The main aim identified was to consider whether physical activity positively impacts psychosocial and emotional mental health and well-being outcomes of underprivileged children and adolescents.

Objectives: Three objectives were identified: (i) To identify mechanisms that explain why there is an association; (ii) To identify factors which influence the association; (iii) To explore whether time spent in an activity or its intensity influence the association.

## 2. Materials and Methods

### 2.1. Design and Eligibility Criteria

The integrative review design identified by Whittemore and Knafl [39] has five stages; (a) identifying the problem; (b) searching the literature; (c) data evaluation; (d) data analysis and (e) presentation of findings. The integrative review is the methodology of choice because, unlike systematic reviews and meta-analyses, it combines experimental and non-experimental data, thus enhancing the depth and breadth of the results, whilst also explaining the mechanisms which determine a particular outcome [39,40]. The conclusions drawn from a comprehensive review of this nature will present the state of the field and will provide evidence from which to design and implement future policy and practice. For the purpose of this review, the philosophical position that is assumed is that of a pragmatist which has no fixed ontological or epistemological stance. Rather, the aims of this project were designed in order to achieve usable conclusions from research.

The PRISMA flow diagram [41] was used to aid and represent the search process and output. The search processes were developed according to the Cochrane Reviewer’s Handbook for Systematic Reviews [42], whilst also being sensitive to qualitative research [40].

The search concept tool, SPIDER [43], was used to determine the search criteria as follows:Sample—The specified sample was studies which involved children and adolescents aged between 5 and 18 years, specifically disadvantaged individuals. The reason for focussing on underprivileged children is that this population group are at greater risk of negative mental health outcomes, than their more privileged peers [44,45]. To ensure there was consistency with the sample, studies which included adults and children under the age of 5 were excluded, unless the data sets were analysed independently. In this instance, only the results relating to children over 5 years of age and adolescents were used. Only studies which included group-based PA were included.Phenomenon of Interest—This review aimed to explore the link between physical activity (PA) and positive mental health outcomes in disadvantaged children and adolescents, whilst also revealing associations and mechanisms that led to the outcomes. Mental health encompasses a range of different outcomes, including, self-esteem; self-worth/self-concept; confidence/competence; autonomy; emotional regulation; anxiety; depression and peer attachments/social skills [46]. These outcomes can be divided into three distinct groups: social, internal, and physical aspects of mental health. Any study which included a reliable measure of one or more of these outcomes was included in the review. The main focus was on studies that considered if **PA positively impacts psychosocial and emotional mental health outcomes of underprivileged children and adolescents**; however, studies which also considered other factors were included providing there was sufficient data relating to at least one of the mental health outcomes outlined above. This allowed the research to address a number of questions. Is the association influenced by the type of activity or its duration and intensity? Aside from time and intensity, are there any other common factors which seem to influence the association? What are the underlying biopsychosocial, political or environmental mechanisms which explain why there is an association?Design—In order to ensure there was depth and breadth to this review, the included studies were not limited to a single methodology. Various qualitative approaches were used, for example, action research, types of grounded theory, descriptive interpretative designs, or phenomenology. Due to the accumulation of evidence, single person case studies were excluded, along with studies which made use of fictional stories.Evaluation—Studies which used age-relevant questionnaires, such as the ‘Piers–Harris Children’s Self-Concept Scale—second edition’ [47], Kidscreen-27 [48] or the ‘Strengths and Difficulties Questionnaire’ [49], were included if used in relation to the impact of physical activity on mental health. Surveys and focus groups were also included if they focussed on the phenomena of interest. Specific areas of interest included the type, duration, and intensity of the physical activity and whether any of these factors were more impactful on positive mental health outcomes. Variations due to gender were also explored, where data allowed.Research Type—Qualitative, quantitative, and mixed methods research were included.Other Criteria—Due to the worldwide concern that children are leading more sedentary lives [9] and that mental health problems are on the rise [50] with a disproportionate impact on underprivileged children [45], there has been an increase in research conducted in this area in recent times. However, it is still a relatively new area of research; therefore, to provide a clear picture of what is currently known about the relationship between PA and the mental health of disadvantaged children and adolescents, it was felt that all relevant studies should be included with no date restrictions.

An electronic search was undertaken within several databases from inception until May 2020 in the following databases; Medline, PubMed, CINHAL, SportsDiscus, PsychInfo. Key search terms can be seen in Table 1. This meets guidance requirements of using a diverse range of search terms [42]. Further searching was undertaken within other electronic search engines including Google Scholar and ScienceDirect, as well as citation chasing and searching for grey literature.

### 2.2. Study Selection

Process for selecting studies:The studies found through each database or search method were initially screened by title. Any studies which did not have a focus on positive mental health outcomes in relation to physical activity were excluded, along with those which used adults as the sample.Duplicates were removed.The next stage of screening involved reading the abstracts of each study. Only relevant studies with a positive mental health outcome measure were retained.Finally, upon reading the full text articles, any which did not meet the eligibility criteria, i.e., used the incorrect sample, were excluded.At stage 1 and stage 3 of this process, author AS also screened the articles for quality purposes and no conflicts occurred.The Mixed Methods Appraisal Tool version 2018 (MMAT) was used to critically appraise all included studies [51]. The 2018 version of the MMAT is the third to be developed since its inception. It consists of a checklist with 25 items and 2 screening questions and can be used for the appraisal of five different study designs: qualitative; quantitative descriptive; randomised control; non-randomised control and mixed methods [51]. Alterations have been made to it over time following studies of its effectiveness, thus enhancing the validity and reliability of the tool [52,53,54]. In order to provide a comprehensive appraisal of the qualitative studies, additional criteria were taken from the COREQ checklist [55] and added to the MMAT [51].

### 2.3. Data Analysis

There are four stages of data analysis outlined by Whittemore and Knafl [39]. Stage 1 is data reduction which requires the development of an organised system and framework to classify, extract and then code the data. The data obtained by this review was initially divided by type of evidence: qualitative, quantitative, and mixed methods and then further split into subgroups: type of physical activity and specific mental health outcome. All relevant data needed to be collected, coded, and formulated into a simple framework, using pre-determined, reliable measures [39]. The data was extracted by LR, as lead researcher, using a quantitative-led approach as it is the most prevalent type of data, in three stages: data immersion; tabulation and coding [56].

Stage 2 is data display and involves the combination and representation of data relating to specific subgroups from multiple sources [39]. The data was tabulated with the following column headings: study; methodology, aim and eligibility; participants, sampling and setting; key intervention details and data collection methods, focus and analysis. Author AS conducted random checks on the use of the MMAT and the extracted content to ensure all relevant data were collected and recorded correctly. CR acted as moderator, so that as a team any conflicts could be resolved for quality assurance purposes. However, there were no conflicts.

Stage 3, data comparison, requires the examination of the data displays developed from primary sources, in an attempt to establish patterns or themes [39]. Concept mapping [56] was used to explore the relationships between the key concepts of physical activity and positive mental health outcomes.

Stage 4, conclusion drawing and verification, involves the development and presentation of conclusions based on the patterns, commonalities and differences that were identified during the display and comparison stages of analysis [39]. The data analysis audit trail can be found in the Appendix A.

## 3. Results

### 3.1. Systematic Search and Study Selection

Figure 1 shows the process for selecting the studies and is adapted from the Prisma Flow Diagram [41].

### 3.2. Demographics Summary

Sixteen studies were included in this integrative review: two qualitative, twelve quantitative (cohort, *n* = 6; cross-sectional, *n* = 2; RCT, *n* = 4) and 2 mixed methods (cohort, *n* = 1, RCT, *n* = 1). A total of 5886 children and adolescents aged between 8 and 16 years participated in the studies. There were 3180 boys, 2581 girls and 125 not identified from one study [57].

In addition, one study [36] included 13 parents and 4 personnel who were involved in the programme delivery alongside the adolescent participants, whilst another [37] gathered data exclusively from 10 parents whose children had completed the intervention. The studies were completed in ten different countries with six in the USA, two in Australia and one in each of the UK, Columbia, Chile, Ireland, Israel, India, South Africa and Canada.

The type of physical activity/sport varied considerably across the studies with: multi-sports (*n* = 4); unspecified (*n* = 3); yoga (*n* = 4); martial arts/group sports combined (*n* = 1); dance (*n* = 1); aerobic exercise (*n* = 1); low-risk boxing (*n* = 1) and sport focussed specifically on improving muscular fitness (*n* = 1). The intensity of activity ranged from moderate to vigorous, the duration from 30 min to six hours and the frequency from daily to weekly. Only group PA was included and most was non-competitive (*n* = 8), one study used competitive PA, one incorporated a mix of competitive and non-competitive PA, and the remainder did not specify whether the PA was competitive or non-competitive (*n* = 6).

Across the studies, different mental health outcomes were measured, including depression (*n* = 1) and self-esteem (*n* = 1). Three studies combined self-esteem, anxiety, and depression, whilst another combined anxiety, depression, and aggression. Six studies looked holistically at psychological wellbeing and included factors such as self-worth, self-concept, self-esteem, and peer attachments. Two studies measured socio-emotional wellness, including aggression, self-control, and emotional regulation. The final three studies used a Positive Youth Development setting [58] to develop and measure changes in mental health outcomes such as self-worth, social competence, and self-control.

Appendix A provides a more detailed description of the demographics and characteristics of all included studies.

### 3.3. Quality Assessment Summary

Table 2, Table 3, Table 4 and Table 5 provide a detailed assessment of the quality of included articles. The full between and within study quality analysis can be found within the Appendix A.

### 3.4. Overview of Results

#### 3.4.1. Internal Aspects of Mental Health

Twelve (12/14; 86%) studies which explored internal aspects of mental health saw improvement in one or more outcomes [29,30,32,35,38,57,61,62,63,64,65,67]. Two studies did not show any significant changes [31,66]. The full results can be found in Appendix A.

##### Sub-Theme 1: Depression

Depression was measured four times [38,61,62,63] but the results varied. Cross-sectional evidence identified a strong association (OR = 0.89, 95%CI, 0.85–0.93) between an increase in self-reported PA and a reduced chance of depression in adolescents [63]. Specifically, one extra hour of PA per week, corresponded to an 11% lower likelihood of depression.

Three intervention studies considered depression, one [61] showed significantly decreased levels of depression post-intervention (d = 0.74), resulting in a moderate overall effect on depression levels in the 9–10 year-old experimental participants. The second [57] showed a reduction in the depression levels of the adolescents in the experimental group; however. the baseline levels of depression were not comparable between the experimental and control groups and this had a moderate effect on the findings (d = 0.53), meaning there was no significant overall effect. The third [62] also involved adolescents and reported no significant overall effect on depression levels when comparing the experimental to the control groups (score difference = −0.07, 95% CI −0.83–0.69, *p* = 0.436).

Although the overall results for Velásquez et al.’s study [57] were affected by the baseline differences there were some clear findings. The depression levels in the experimental participants were significantly impacted by attendance (λ = 0.87, *p* < 0.05), age (λ = 0.96, *p* < 0.02) and gender (λ = 0.91, *p* < 0.05). As expected, the children who attended more frequently experienced a bigger decrease in depression levels following the intervention than those who attended less frequently. This supports associations identified by Rothon et al. [63]. Whilst the younger children experienced a positive change, there was no significant decrease in the depression levels of the older children [57], which supports Bonhauser et al.’s [62] findings. Gender also proved to be a determining factor, the depression decrease for the boys was statistically significant, but not for the girls [57]. Interestingly, Rothon et al. [63] also found that higher MVPA had a greater impact on depression in boys than girls, as the likelihood of depression was nearly twice as high in the girls (OR = 1.82, 95% CI 1.49–2.24). Crews et al. [61] determined that improvements in cardiovascular fitness (CVF) are a possible mechanism for decreased depression; however, the strength of the association was weakened by the differing group levels of baseline depression.

##### Summary

In summary, the results suggest that higher levels of MVPA correspond with lower levels of depression in children [57,61,63]. However, it should be noted that all three of these studies were assessed to be poor quality, with scores of 2/5. As there was no contradictory evidence found in this review, and the findings were consistent across three studies, it can be concluded with reasonable confidence that higher PA reduces depression level and likelihood in children. The results relating to adolescents are less conclusive. There is some evidence to suggest that higher levels of MVPA do not positively impact depression levels in adolescents [57,62]. As only one study was scored as high quality [62], there is insufficient evidence to conclude with confidence and so further research with adolescents is required. Although, it appears that gender may be a moderator for depression [57,63], again there is insufficient evidence to confirm this and so this needs further exploration too.

##### Sub-Theme 2: Anxiety

Anxiety was measured three times [57,61,62], with significant results in 2/3 (66.6%) intervention-based studies [57,62]. One study [62] showed a significant reduction of 13.7% in anxiety levels for the adolescents in the mixed sport intervention group, as compared to a small reduction of 2.8% for the control group, who took part in a standard PE lesson once, rather than three times, per week (−0.94, 95% CI −1.43 to −0.45, *p* < 0.001). Velásquez et al. [57] also reported a significant, albeit small, decrease in the yoga group’s anxiety levels post-intervention, when compared to the control group (F = 3.87, *p* < 0.05, d = 0.21). In contrast, Crews et al. [61], found no significant difference in anxiety levels following the intervention (F= 0.40, *p* > 0.05, ES = 0.10). However, the anxiety levels at baseline were not comparable, thus affecting the reliability of this study’s findings. Velásquez et al. [57] also collected qualitative data, with participants describing a reduction in stress and an increased ability to relax in all parts of their lives following the yoga intervention, thus supporting the quantitative evidence collected by Velásquez et al. [57] and Bonhauser et al. [62].

##### Summary

MVPA interventions appear to have a positive impact on anxiety levels in both children and adolescents. However, the findings are limited as only three of the included quantitative studies examined anxiety and the type, duration and frequency of the PA intervention varied. It is important to note that only one of the three studies was graded as high quality [62], with the other two poor quality [57,61]; therefore, more research is required to confirm the association.

##### Sub-Theme 3: Self-Esteem

Self-esteem was measured in three intervention studies with 3/3 (100%) showing a significant increase in self-esteem levels post-intervention [61,62,64]. One study [64] involved a single group with no control and found a large and significant overall increase of 9.04% (*p* > 0001) in the self-esteem levels of the female adolescent participants. Bonhauser et al. [62] also showed an overall small but significant increase of 2.3% in the self-esteem of the intervention participants, in contrast to a small decrease for the control group of 0.1% (Score difference − 7.17, 95% CI 4.43–9.91, *p* < 0.001, calculation of effect size not possible). Crews et al. [61] found that there was a significant increase in self-esteem for the experimental group and a large effect size when this was compared with the control group (ES = 0.94). This association was supported by the qualitative results of two studies [36,37] in which the majority of the participants and their parents reported an increase in self-esteem and self-confidence following the intervention.

In one study [61], cardiovascular fitness (CVF) was measured and improved significantly in the aerobic exercise group (ES = 0.66) and it is possible that this may be a mechanism for increased self-esteem, but more research is required for confirmation.

##### Summary

As 2/3 (66%) of the studies were scored as good quality, the results were conclusive, and were supported by the findings of the two qualitative studies [36,37], it can be stated with some confidence that MVPA positively impacts self-esteem. However, more research is required to determine whether age, gender or any other underlying mechanisms influence the association with self-esteem.

##### Sub-Theme 4: Emotional Regulation

Emotional regulation groups together multiple outcome measures: self-control; aggression; coping skills and negative behaviours. Five intervention studies examined one or more of these outcomes [32,57,65,66,67] and there were significant positive changes in 4/5 (80%) studies [32,57,65,67].

##### Sub-Factor 1–Self-Control

One study [65] reported a significant increase in self-control (η*p*^2^ = 0.012, *p* < 0.01).

##### Sub-Factor 2–Emotional Regulation

Shachar et al. [65] reported a significant reduction in negative emotions (η*p*^2^ = 0.026, *p* < 0.001), anger (η*p*^2^ = 0.029, *p* < 0.001), hostility (η*p*^2^ = 0.026, *p* < 0.001) and physical aggression (η*p*^2^ = 0.065, *p* < 0.001). Small effect sizes were identified across findings. In one study [57], there was a significant decrease in aggression levels observed post-intervention (d = 0.14) for the yoga group. Frank et al. [32] identified significant positive improvement in emotional regulation (*p* = 0.05, d = 0.12). Only one study [66] did not report any positive findings for emotional regulation. Self-control was measured and there was a small but insignificant increase. Velásquez et al. [57] also collected qualitative data in which high school students described having better control of their emotions and being able to calm down more easily. Similarly, the participants in Berger et al.’s study [67] reported feeling calmer, being better behaved, and also using yoga to control their emotions when angry. These qualitative findings provide additional support for the quantitative evidence.

##### Sub-factor 3–Interaction Effect Between Self-Control and Emotional Regulation

The results obtained by Shachar et al. [65] showed that gender moderated the effect of the intervention on emotional regulation. In both boys and girls, it was found that self-control was positively correlated with positive emotions (girls: r = 0.26, *p* < 0.01; boys: r = 0.23, *p* < 0.01). In the girls it was negatively correlated with negative emotions (r = 0.16, *p* < 0.01); hostility (r = 0.12; *p* < 0.05) and physical aggression (r = 0.13, *p* < 0.05). In the boys, there was also a negative correlation with negative emotions (r = 0.26, *p* < 0.01), physical aggression (r = 0.24, *p* < 0.01) and, in contrast to the girls, anger (r = 0.22, *p* < 0.01). The effect size was small but significant for all correlations. There was a significant direct link between changes in self-control skills and physical aggression in the girls (β= −0.29, *p* = 0.012). Contrastingly, in the boys there was a significant direct link between hostility and physical aggression (β = 0.62, *p* = 0.003) and an indirect link between changes in self-control and physical aggression (95% CI = −0.247, −0.045, *p* = 0.004). As there was a direct link for girls but an indirect link for boys between self-control skills and aggression, further research is required to determine whether the underlying mechanism differs.

##### Sub-Factor 4–Negative Behaviours

One study [67] found a significant decrease in negative behaviours following the yoga intervention (d = 0.40). Another study [65] reported significantly increased teacher-rated negative behaviours (η*p*^2^ = 0.011, *p* < 0.01) with a small effect size. Frank et al. [32] stated that no significant effect was observed for ‘attitudes towards violence’.

##### Sub-Factor 5–Coping

Frank et al.’s study [32] also involved a yoga intervention and found significant positive improvements in primary coping, (*p* = 0.02, d = 0.15), secondary coping (*p* = 0.01, d = 0.14) and positive thinking (*p* = 0.05, d = 0.13).

##### Summary

As 4/5 (80%) of the included studies which examined emotional regulation found positive results, and this was corroborated by the qualitative evidence [57,67], enhanced MVPA appears to impact emotional regulation in children and adolescents aged 9–16 years. However, only one study was good quality [65], one was moderate quality [66] and three were poor quality [32,57,67].

Gender certainly appears to moderate the results and the fact that underlying mechanisms for the link between self-control skills and aggression may differ between boys and girls is interesting, but as the evidence is limited, caution is required when drawing conclusions.

As 3/5 of the included studies used yoga [32,57,67] and 3/3 (100%) produced significant results, it can be concluded with relative confidence that yoga positively impacts emotional regulation in children and adolescents.

##### Sub-Theme 5: Psychological Wellbeing

Psychological wellbeing was measured in two cross-sectional studies [29,38] and one intervention study [35] with significant results in 3/3 (100%). Both cross-sectional studies used the Kidscreen-27 questionnaire [48] in which psychological wellbeing is determined by a combination of positive emotions, emotional regulation, and life satisfaction measures. Psychological wellbeing was significantly higher in children and adolescents who participated in higher levels of MVPA regularly (2: *p* ≤ 0.01, η*p*^2^ 0.02; 12: *p* < 0.001, η^2^ = 0.016). Salvini et al. [38] also observed that psychological wellbeing was positively correlated with MVPA (*p* < 0.001, η^2^ = 0.016). The intervention study [35] used flourishing as the outcome measure of psychological wellbeing and found that there was a small but significant enhancement following the intervention (d = 0.35).

Cross-sectional evidence [29,38] identified volume and frequency of MVPA as a mechanism for improved psychological wellbeing, with Salvini et al. [38] showing that higher levels of MVPA had a significant effect, regardless of age, gender, BMI, CRF or SES. Four potential mechanisms for the change in psychological wellbeing were identified by Lubans et al. [35]: autonomy choice (A = 0.15, SE = 0.06, *p* = 0.016); screen time (A = −0.21, SE = 0.06, *p* < 0.001); muscular fitness (A = 0.012, SE = 0.05, *p* = 0.011) and resistance training skill competency (A = 0.48, SE = 0.04, *p* < 0.001). Of these, three were statistically significant as mediators: an increase in autonomy choice (AB = 0.024, 95% CI = 0.004–0.052), a decrease in screen time (AB = 0.038, 95% CI = 0.007–0.080) and increased muscular fitness (AB = 0.026, 95% CI = 0.003–0.061).

##### Summary

As 3/3 studies (100%) that examined psychological wellbeing as a mental health outcome found significant results and that increased MVPA was a mediator, it can be concluded with reasonable confidence that higher levels of MVPA have a significant effect on psychological wellbeing. It should be noted that the studies varied in quality from poor [29] to moderate [35] to good [38], and so additional high-quality research in this area would be useful to reinforce the results. Increased autonomy choice and muscular fitness, and decreased computer time were identified as mediators [35]. However, as only one study looked at these factors, caution is required when drawing conclusions.

##### Sub-Theme 6: Self-Worth

Self-worth was measured in two intervention studies [30,67], with a significant increase in 1/2 (50%) studies. Ullrich-French et al. [30] found a moderate significant increase in global self-worth (F = 17.12, η^2^ = 0.15, *p* < 0.01) and a small but significant increase in physical self-worth (F = 11.12, η^2^ = 0.05, *p* < 0.01). Contrastingly, Berger et al. [67] recorded no significant difference in global self-worth post intervention.

Age moderated the findings for self-worth as only the adolescents showed a significant increase in physical self-worth, whilst there was no significant change for the 9–11 year-old participants [30]. Interestingly, an increase in physical competence was found to predict the increase in physical self-worth and global self-worth.

##### Summary

One [30] was a good quality cohort study, so some value can be placed on the findings from this study. The other was a poor-quality study [67]. Therefore, the findings for self-worth should be noted but no firm conclusions can be drawn based on these two studies alone.

##### Sub-Theme 7: Motivation Towards Physical Activity

Motivation towards physical activity was measured in one intervention study [30] and was discussed in one qualitative study [36]. Ullrich-French et al. [30] recorded no significant overall effect; however, when split by age, a small but significant overall increase in motivation towards physical activity was found for the 9–11 years group (F = 7.46, η^2^ = 0.07, *p* < 0.01). There was no significant effect for the 12–16 year olds. Beaulac et al.’s [36] study was qualitative and examined the effect of a dance intervention. Most of the participants and some parents reported an increase in motivation to participate in additional physical activity outside of the organised dance lessons.

##### Summary

There is insufficient evidence to suggest a link between an enhanced MVPA programme and increased motivation towards physical activity, especially as there was no overall effect in the good quality study [30] and the quality assessment of the second study [34] found major concerns.

#### 3.4.2. Social Aspects of Mental Health

Six studies (6/6; 100%) which explored social aspects of mental health, saw improvement in one or more of the measured outcomes [29,30,35,38,57,66]. The full results can be found in Appendix A.

##### Sub-Factor 1: Social Competence

Social competence was measured in two intervention studies [30,66] with differing results. Ullrich-French et al. [30] recorded a small but significant increase in overall social competence (F =15.67, η^2^ = 0.07, *p* < 0.01), but contrastingly, Anderson-Butcher et al. [66] found no significant change in social competence post-intervention.

Although Ullrich-French et al. [30] found a significant effect overall, when examined more closely the effect was found to be moderated by gender, with only girls experiencing a significant effect of moderate magnitude (F = 12.78, η^2^ = 0.11, *p* < 0.01). It was also determined that an increase in social competence predicted an increase in hope (r = 0.45), whilst an increase in both social competence and leader support predicted an increase in global self-worth (r = 0.52) and motivation towards physical activity (r = 0.64). Interestingly, although Anderson-Butcher et al. [66] did not record a significant increase in social competence, a small increase was observed and was predicted by a significant increase in teamwork (r = 0.46) and sense of belonging (r = 0.56).

##### Summary

One study was graded as a good quality study [30] and the other [66] was graded as moderate quality. As only two studies included social competence as an outcome measure and the results differed, more research is required to form any conclusions. It appears that there may be a link between an increase in social competence, and increases in other factors, namely: global self-worth; hope; motivation towards physical activity; teamwork and sense of belonging. However, as only two studies included social competence as an outcome measure and the results differed, no firm conclusion can be drawn.

##### Sub-Factor 2: Social Responsibility

Two intervention studies [57,66] measured social responsibility and both (2/2; 100%) found a significant effect following the intervention. The effect size was moderate in both studies ([57] r = 0.41, *p* < 0.05; [66] d = 0.05). These results were supported by the qualitative findings of Riley and Anderson-Butcher’s study [37] in which 10/10 (100%) of the parent participants described positive changes in their children’s social skills and responsibility towards others.

It was observed by Velásquez et al. [57] that the change in social responsibility following the intervention was impacted by age (λ = 0.97), with a significant effect observed in adolescents only. Similarly, Anderson-Butcher et al. [66] found that age predicted the change in social responsibility (r = 0.14, *p* < 0.05), along with sense of belonging (r = 0.43, *p* < 0.05).

##### Summary

As there were two quantitative and one qualitative study which produced significant results, it can be concluded with some confidence that there is an association between enhanced MVPA and social responsibility. It looks as though age is a moderator for changes in social responsibility; however, there are only two studies to draw results from, one of which was moderate quality [66] and the other was poor quality [57], so there is insufficient data to draw a conclusion.

##### Sub-Factor 3: Support from Others

‘Support from others’ includes three measures: autonomy and parental support; social support and leader support. It was measured in two cross-sectional [29,38] and two intervention studies [30,35] with a significant overall increase in 3/4 (75%) studies. Cross-sectional evidence found that both autonomy and parental support (2: F = 6.10, *p* = 0.01, η*p*^2^ = 0.11; 12: F = 12.11, *p* < 0.001, η^2^ = 0.029) and social support (2: F = 4.60, *p* ≤ 0.03, η*p*^2^ = 0.01; 12: F = 6.56, *p* < 0.001, η^2^ = 0.016) were significantly higher for children who met or exceeded recommended MVPA levels. Qualitative evidence [36] provides additional support for an association between enhanced MVPA and parent and social relations, with many of the participants stating that their relationships with both their parents and peers improved following the intervention.

In one study [35], there was a small but significant effect on autonomy choice post-intervention (A = −0.21, SE = 0.6, *p* = 0.16). This supports the association found by Breslin et al. [29] and Salvini et al. [38]. One study [30] measured leader support and found no significant change overall.

Autonomy choice was found to act as a mediator for overall enhanced wellbeing (AB = 0.026, 95% CI = 0.004–0.055) [35]. However, there were various other factors included in this intervention which were also found to enhance wellbeing, such as reduced computer time and improved muscular fitness; therefore, a causal link cannot be applied to increased autonomy and enhanced wellbeing from these results alone. Interestingly, although Ullrich-French et al. [30] found there was no significant effect on leader support overall, there was a small but significant effect on leader support for the children aged 9–11 years (F = 2.64, η^2^ = 0.03, *p* < 0.05) so age mediated the results. One study [29] reported gender as a moderator, as the results for autonomy and parent support (F = 7.43, *p* < 0.01, η*p*^2^ = 0.14) and social support (F = 6.36, *p* < 0.01, η*p*^2^ = 0.12) were significantly higher for girls than for boys.

##### Summary

As 3/4 (75%) studies found that increased and enhanced MVPA had a significant impact on autonomy, and only one was graded as poor [29], with the others moderate [35] and good [38] quality studies, there is sufficient evidence to conclude with reasonable confidence that higher MVPA levels improve autonomy. As both cross-sectional studies [29,38] also found there was a significant effect on social support and parental support, and this was supported by qualitative evidence [36], it is likely that increased MVPA has a positive effect on those factors too, although for a firm conclusion, further studies should be examined. No conclusions can be drawn in relation to age and gender as moderators; however, there is some limited evidence for a link, so further work is required here too.

#### 3.4.3. Physical Aspects of Mental Health

Five studies [29,30,38,66,67] explored physical aspects of mental health and of these, 3/5 (60%) saw a significant improvement in one or more of the measured outcomes [29,30,38]. The full results can be found in Appendix A.

##### Sub-Factor 1: Physical Wellbeing

Physical wellbeing was measured in two cross-sectional studies [29,38] and a small but significant increase was found in both studies (2/2, 100%) for the children who met or exceeded recommended MVPA levels ([29]: F = 31.77, *p* ≤ 0.01, η*p*^2^ = 0.06; [38]: F = 15.89, *p* < 0.001, η^2^ = 0.037). This association was supported by the qualitative findings of study 7, in which most participants described increases in fitness, strength and energy after the intervention.

##### Summary

Although a significant effect was observed in both studies, only one was good quality [38] and the other was poor quality [29], and the findings of just two cross-sectional studies is not sufficient to draw conclusions.

##### Sub-Factor 2: Physical Competence

Physical competence was measured in two intervention studies [30,66], with 1/2 (50%) producing significant results [30]. One study [30,31] showed a small but significant effect of MVPA on physical competence (F = 18.41, η^2^ = 0.09, *p* < 0.01) but there was no significant effect reported in the other [66].

Both studies used a sample of 9–16 year-olds and both were conducted in holiday camps, based around positive youth development. Strong positive correlations were found between physical competence and physical self-worth (r = 0.72) and attraction to physical activity (r = 0.73), whilst there was a moderate positive correlation with self-worth (r = 0.62) and hope (r = 0.44). This association between an MVPA intervention and physical competence was a finding that emerged from the two qualitative studies [36,37], with both participants and parents noting a significant increase in self-belief in the physical activity context.

##### Summary

Clearly there is insufficient evidence to draw firm conclusions relating to physical competence and more work is needed. However, the correlations with other outcomes suggest there is an association and were found in a good quality study [30], so this warrants further examination.

##### Sub-Factor 3: Physical Self-Worth

Physical self-worth was measured in two intervention studies [30,67], with 1/2 (50%) producing significant results [30]. One study [30] found a small but significant overall increase in physical self-worth following the intervention (F = 11.12, η^2^ = 0.05, *p* < 0.01) but there was no significant effect observed in the other [67].

The results were moderated by age as there was a fairly large and significant effect on those aged 12–16 years, whilst there was no significant change observed in the 9–11 year olds (F = 17.26, η^2^ = 0.14, *p* < 0.01) [30]. This study [30] also displayed a strong positive correlation with self-worth (r = 0.72) and attraction to physical activity (r = 0.73) and a moderate positive correlation with hope (r = 0.46).

##### Summary

Again, there is insufficient evidence to form conclusions regarding the effect of enhanced MVPA on physical self-worth. However, the correlations are interesting, were reported in a good quality study and warrant further investigation.

## 4. Discussion

This was the first study to provide an integrative understanding of the impact of physical activity on psychosocial and emotional mental health outcomes in underprivileged children and adolescents. The main findings showed that MVPA significantly impacts the mental health of disadvantaged children and adolescents with 15/16 (93.8%) of the included studies recording a significant positive change in at least one internal, social, or physical outcome. Only one study [31] did not record a significant improvement, but equally the study was not found to be detrimental to the mental wellbeing of the participants. The findings were not unexpected as previous research of children and adolescents in the general population had produced similar results in relation to MVPA and improvements in mental health outcomes, e.g., [16,18]. The focus of this discussion is on the mechanisms and influences which explain the association between MVPA and positive mental health outcomes.

### 4.1. Mechanisms and Influencing Factors

The current review identified that autonomy support was a possible mechanism across intervention-based research and cross-sectional studies for improved mental health and wellbeing outcomes. Previous research has reported that children and adolescents with higher levels of perceived autonomy support, spend more time participating in physical activity than their peers [68,69] thus supporting the theory of self-determination [70], which states that autonomy support enhances intrinsic motivation. The findings from this review, when taken in conjunction with previous research, show that where physical activity is delivered in such a way that it enhances autonomy support, the benefits are twofold. The increase in the children’s and adolescents’ perceived autonomy support acts as a mechanism for improved psychological wellbeing, whilst also leading to more time spent in MVPA due to its positive impact on intrinsic motivation. Future research should explore MVPA interventions based around autonomy support to determine whether it is also a mechanism for improvements in other social, internal, and physical mental health outcomes.

There were two key influencing factors identified across the studies, age [30] and gender [29,63]. Although some studies reported a positive impact of MVPA on the mental health of both underprivileged children and adolescents, the results for some outcomes proved significant only for adolescents. This supports the findings from previous studies of children and adolescents within the general population [18]. To the author’s knowledge, no research has been conducted into gender as an influencing factor, when linking MVPA to positive mental health outcomes in youth.

Ullrich-French et al. [30] reported a significant increase in physical self-worth and motivation in the adolescent participants post-intervention but no significant effect for the younger children. The result for physical self-worth is interesting as previous research has found that MVPA positively impacts physical self-worth in both children and adolescents within the general population [71,72]. Further research of MVPA on physical self-worth is needed with children and adolescents of all ages, considering both disadvantaged and more privileged individuals, to determine whether age and socioeconomic status influence the effect.

The findings from this review suggest that gender is an influencing factor for a range of mental health outcomes. Interestingly, in two studies [57,65], the girls recorded a significantly greater reduction in depression levels with increased PA than the boys. Previous research is limited regarding the impact of MVPA on depression in children and adolescents of both genders, and to the authors’ knowledge, no prior studies have explored whether the effect differs for boys and girls; therefore, it is not possible to make a causal statement. However, as the findings of two studies in this review agree, the link between depression and MVPA for boys and girls from all socioeconomic backgrounds warrants further separate investigation.

Ullrich-French et al. [30] recorded a significant increase in social competence following the intervention but, when separated by gender there was a large effect for the girls and no change for the boys. The change in social competence was correlated with positive changes in physical competence, self-worth, motivation, and hope, suggesting that an improvement in one outcome may act as a mechanism for improvement in others. Shachar et al. [65] recorded an increase in self-control for both genders; however, whilst there was a direct link between self-control and aggression in the girls, there was an indirect link involving hostility for the boys. These findings suggest that the underlying mechanisms may differ for boys and girls. A better understanding of these mechanisms would allow for the development of effective interventions, specific to each gender for greater improvements in the mental health outcomes of disadvantaged children and adolescents. As there is no literature available for comparison, more studies are required to develop a greater understanding of the gender differences.

### 4.2. Limitations

There were a number of limitations with this review: (a) there is currently a low number of studies considering outcomes and mechanisms identified by the review and further fully powered research is required; (b) most of the available studies only used adolescents, with just 5/16 (31.3%) using children under the age of eleven; (c) only one study considered longer term outcomes at two years and further research is needed to support these findings; (d) yoga was the only specific MVPA type which had sufficient studies to determine its effectiveness and insufficient studies have been completed with other specific PA types or sports to draw conclusions regarding their effectiveness; and (e) there is currently insufficient data available to determine if MVPA, in all of its forms, positively impacts every outcome. It is acknowledged that the limited number and heterogeneity of the included studies is a significant limitation which impacts the generalizability of the results.

### 4.3. Implications

This review has shown that increased or enhanced MVPA positively impacts the mental health of underprivileged children and adolescents. These results should be used to encourage schools to enhance their PE provision and to increase, not decrease, the offering during adolescence, as MVPA is particularly beneficial for this age group. As schools are not the only setting in which young people can participate in sport or PA, sport governing bodies and local councils should target low socioeconomic areas and ensure that underprivileged youth are given access to a wide offering of sports and activities.

Autonomy support was found to be a key mechanism for positive change; therefore, it is important that all organised MVPA is taught in an autonomy supportive environment to maximise the impact on young people’s mental health and wellbeing. The findings from this review alone are insufficient to fully understand the gender differences, so it is important that future research separates gender when considering the impact of MVPA on specific mental health outcomes. Within this research, an attempt should be made to identify the underlying mechanisms for change, as the findings from this review suggest that these may differ for boys and girls. This could have implications for the delivery of organised MVPA for both genders. As age was also identified as an influencing factor and the results for children and adolescents differed across a range of mental health outcomes, additional studies with both age groups are necessary to determine the reasons behind the effect. These findings could also have implications for the delivery of MVPA, based on age.

## 5. Conclusions

It is well known that disadvantaged children and adolescents are at increased risk of developing mental health problems. This review was both timely and necessary to develop a better understanding of the positive impact physical activity has on psychosocial and emotional mental health outcomes in this particular population. Importantly, a strong association was identified between an autonomy supportive PA setting and an enhanced impact on positive mental health outcomes. By drawing together and integrating all past research, this review has both confirmed the links between MVPA, autonomy support and positive mental health outcomes, and identified the gaps in current knowledge, thus providing a direction for future research.

## Figures and Tables

**Figure 1 behavsci-10-00171-f001:**
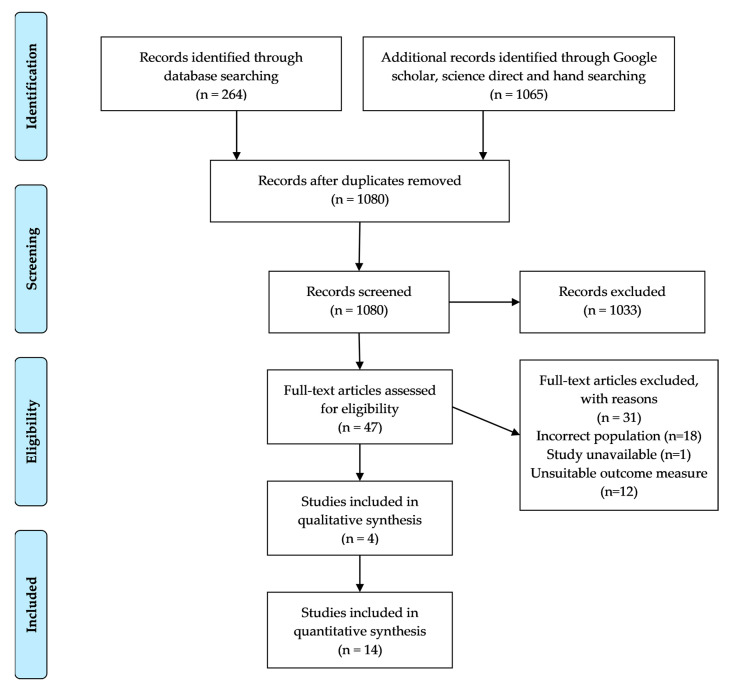
PRISMA Flow Diagram adapted from Moher [41].

**Table 1 behavsci-10-00171-t001:** Search Terms Using SPIDER Acronym.

SPIDER Tool	Search Terms
S—Sample	(Child * OR adolescent * OR youth OR juvenile) AND (underprivileged OR disadvantaged OR “low socioeconomic” OR socioeconomic OR “socioeconomic status”)
P of I—Phenomenon of Interest	(“physical activity” OR “physical education” OR PE OR PA OR sport OR tennis OR swimming OR football OR cricket OR gymnastics OR dance OR rugby OR hockey OR netball OR yoga) AND (“mental health” OR wellbeing OR well-being OR “self-esteem” OR “self-worth” OR anxiety OR “self-concept” OR “coping skills” OR “emotional regulation” OR confidence OR resilience)
D—Design	(Question * OR survey OR “focus group” OR interview * OR phenomenology OR grounded theory OR action research OR experiment* OR observ *)
R—Research Type	(Qualitative OR quantitative OR mixed methods)

**Table 2 behavsci-10-00171-t002:** Quality Assessment for Qualitative Studies using MMAT [50] plus additional criteria adapted from COREQ [55].

Authors	Paradigmatic Stance/Methodology	1.1 Qualitative Approach Appropriate?	1.2 Qualitative Data Collection Methods Adequate?	1.3 Findings Adequately Derived from the Data?	1.4 Interpretation of Results Sufficiently Substantiated by Data?	1.5 Coherence Between Qualitative Data Sources, Collection, Analysis, and Interpretation?	1.6 Researcher/Interviewer Training and Experience?	1.7 Researcher Bias Accounted for Satisfactorily?	1.8 Sampling Method Appropriate and Are Dropouts Accounted for?	1.9 Data Collection Appropriate?	Score	Methodological Limitations Based on Munthe-Kaas [59]
Beaulac et al. [36]	Qualitative study with no identification of paradigmatic stance or methodology	Yes (although mixed methods may have provided a greater depth of data)	No (open-ended questionnaires could also be used to give participants a chance to respond anonymously)	Yes	Yes	No (difference sources used for results are not clear)	Cannot tell	No (researcher sometimes involved in the intervention delivery)	No (49% of youths discontinued the program)	Cannot tell (It is not stated whether the data was recorded anonymously)	3/9	Moderate to serious concerns—Flaws with the methodology, including researcher reflexivity, and limitations with the data collection and analysis.
Riley and Anderson-Butcher [37]	Constructivism/ Social constructivist grounded theory	Yes	Yes	Yes	Yes	Yes	Yes	Yes	Yes	Yes	9/9	Very minor concerns—Larger sample size possible.

**Table 3 behavsci-10-00171-t003:** Quality Assessment for quantitative randomised control trials using MMAT [51].

Authors	2.1 Is Randomisation Appropriately Performed?	2.2 Are the Groups Comparable at Baseline?	2.3 Are There Complete Outcome Data?	2.4 Are Outcome Assessors Blinded to the Intervention Provided?	2.5 Did the Participants Adhere to the Assigned Intervention?	Score	Strength of Evidence (Adapted from Anon. [60]	Comments
Crews et al. [61]	Cannot tell	No (imbalance in baseline weight, trait anxiety and depression)	Yes	Cannot tell	Yes	2/5	Level II Evidence Moderately Low Risk of Bias	Poor Quality RCT
Terry et al. [31]	Yes	Yes	Yes	Cannot tell	No (only 50% of the ‘Box’Tag’ programme and 32% of the control ‘Rock and water’ programme attended 15 or more of the 19 sessions)	3/5	Level II Evidence Moderately Low Risk of Bias	Poor Quality RCT
Lubans et al. [35]	Yes	Yes	Yes	No (only at baseline)	Yes	4/5	Level II Evidence Moderately Low Risk of Bias	Moderate Quality RCT
Frank et al. [32]	Cannot tell	Yes	Yes	Cannot tell	Yes	3/5	Level II Evidence Moderately Low Risk of Bias	Poor Quality RCT

**Table 4 behavsci-10-00171-t004:** Quality Assessment for Quantitative Non-Randomised studies using the MMAT [51].

Authors	3.1 Are the Participants Representative of the Target Population?	3.2 Are Measurements Appropriate Regarding Both the Outcome and Intervention (or Exposure)?	3.3 Are There Complete Outcome Data?	3.4 Are the Confounders Accounted for in the Design and Analysis?	3.5 During the Study Period, Is the Intervention Administered as Intended?	Score	Strength of Evidence (Adapted from Anon. [60]	Comments
Bonhauser et al. [62]	Yes	Yes	Yes	Yes	Yes	5/5	Level II Evidence Moderately Low Risk of Bias	Good Quality Cohort study
Ullrich-French et al. [30]	Yes	Yes	Yes	Yes	Yes	5/5	Level II Evidence Moderately Low Risk of Bias	Good Quality Cohort study
Breslin et al. [29]	Yes	No (measurement of MVPA relied on child’s memory and truthfulness)	Yes	No	Yes	3/5	Level IV Evidence High Risk of Bias	Poor Quality Cross-sectional study
Rothon et al. [63]	Yes	No (measurement of MVPA relied on adolescent’s memory and truthfulness)	No	Yes	No	2/5	Level III Evidence Moderately High Risk of Bias	Poor Quality Cohort Study
Sethi et al. [64]	Yes	Yes	Yes	No	Yes	4/5	Level II Evidence Moderately Low Risk of Bias	Good Quality Cohort study
Shachar et al. [65]	Yes	Yes	Bias: drop-out (time constraints; moving; disinterest in sports) and incomplete data	Yes	Yes	4/5	Level II Evidence Moderately Low Risk of Bias	Good Quality Cohort study
Anderson-Butcher et al. [66]	Yes	Yes	Yes	No (cannot find any mention of confounders or methods to control for them)	Yes	4/5	Level III Evidence Moderately High Risk of Bias	Moderate Quality Cohort Study
Salvini et al. [38]	Yes	Yes	Yes	Yes	Yes	5/5	Level III Evidence Moderately High Risk of Bias	Good Quality Cross-Sectional Study

**Table 5 behavsci-10-00171-t005:** Quality Assessment for Mixed Methods Studies using MMAT [51].

Authors	5.1 Is There An Adequate Rationale for Using a Mixed Methods Design to Address the Research Question?	5.2 Are the Different Components of the Study Effectively Integrated to Answer the Research Question?	5.3 Are the Outputs of the Integration of Qualitative and Quantitative Components Adequately Interpreted?	5.4 Are the Divergences and Inconsistencies Between Qualitative and Quantitative Results Adequately Addressed?	5.5 Do the Different Components of the Study Adhere to the Quality Criteria of Each Tradition of the Methods Involved?	Score	Strength of Evidence (Adapted from Anon. [67]	Comments
Velásquez et al. [57]	Yes	No	No	Yes	No (Qual—findings are not adequately derived from the data; Quan—extent of randomisation unclear, blinding bias and not comparable at baseline	2/5	Level II Evidence Moderately Low Risk of Bias	Poor Quality RCT
Berger et al. [67]	Yes	No	No	Yes	No (Qual—Data collection methods are not adequate and insufficient findings are derived; Quan—confounders are not accounted for)	2/5	Level II Evidence Moderately Low Risk of Bias	Poor Quality Cohort Study

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
