# Peer review of "The Positive Impact and Associated Mechanisms of Physical Activity on Mental Health in Underprivileged Children and Adolescents: An Integrative Review"

_behavsci, 2020, doi:10.3390/bs10110171_

Round 1

Reviewer 1 Report

Review

Physical activity and positive mental health outcomes in underprivileged children and adolescents and its determining factors: an integrative review

REVIEWING REPORT

The article meets the objective set out, is well structured, contributes to the field of physical activity and mental health by conducting a review on these issues in disadvantaged children and adolescents. In general terms I liked the way it is written in structure, content and format

Below, I detail in each section some positive considerations and the suggestion of some minor change if it is pertinent.

KEY WORDS

I think it would be better to put only 5 key words: physical activity, mental health, childhood, adolescence and exclusion, 7 are too many.

  1. INTRODUCTION

The sections of the introduction are well organized. But I think you should check the beginning of paragraphs as they are repeated. And in the last two paragraphs you should introduce some more reference.

  1. MATERIALS AND METHOD

This section seems to me to be a strong point of the article. The selection criteria are clearly defined, and the process of selecting the articles for final review. The table on page 4 is very relevant. Perhaps 2.1 and 2.2 should be joined. 2.3 describes very adequately the selection process of the studies, grouped in six steps. The .24 which is the data analysis poses 4 well ordered and organized phases

  1. RESULTS

The content of this section is correct. Figure 2 is very graphic. Table 1 must be reissued in tabular format for each article in vertical format. In the one presented in horizontal format, the information from almost all the studies analyzed is cut out and it takes up 10 pages. In section 3.3. I do not see the need for sections 3.3.1 and 3.3.2, I would dispense with them. The following tables (2,3,4, and 5) review formatting because perhaps two of them can be placed per page. 3.4 presents clearly and conciously the sub themes and sub factors. Table 6 should be reissued in another format, in the one presented that is horizontal, the content analyzed is cut and it takes 12 pages.

  1. DISCUSSION

The content of this section is correct. Perhaps I should expand on the limitations and implications. 

  1. CONCLUSIONS

I think it is appropriate that the conclusions are further expanded as it is a section that hardly provides any argumentation.

REFERENCES

The references used are abundant, relevant, of quality, in journals of impact and closely related to the object of study. But of the 98 sources used, only 23 are after 2015. There are two red marks on the list (5 and 94).

Reviewer 2 Report

This paper is describing an analysis of published research concerning the relation of physical activity in underprivileged youth on their mental health.

Both data about mental health and physical activity are precisely described and also the selected studies are controlled for their quality.

The number of accepted studies are small and cover different ages from 10 to 16 years.

The overall conclusion is that depending on age and gender that moderate to vigorous physical activities are related to a better mental health aspects especially autonomy.

Paper is well written and my suggestion is to make the title more attractive to state that :

Physical activity shows better mental health outcomes in underpreviliged youth etc.

Reviewer 3 Report

The manuscript should be condensed to 20 to 25 pages (total).

This manuscript is 47 pages in length. It is challenging to determine what is valid and as such the key content should be highlighted and the manuscript limited to 20 to 25 pages with the abstract, tables and references.

The abstract is confusing.

Please consider rephrasing "suffers from" or "is suffering" when referring to children who have mental health conditions.  This can be offensive to readers.

The number of articles reviewed are limited and the the findings are not significant.

The findings are weak and the number of studies reviewed was limited (16).

Methods section is well done.

Grammatical errors noted throughout.

Round 2

Reviewer 3 Report

The authors appear to satisfactorily made the requested changes. Minor grammatical errors are evident.